# The Effects of Eggs in a Plant-Based Diet on Oxidative Stress and Inflammation in Metabolic Syndrome

**DOI:** 10.3390/nu14122548

**Published:** 2022-06-19

**Authors:** Minu S. Thomas, Lindsey Huang, Chelsea Garcia, Junichi R. Sakaki, Christopher N. Blesso, Ock K. Chun, Maria Luz Fernandez

**Affiliations:** Department of Nutritional Sciences, University of Connecticut, Storrs, CT 06269, USA; minu.thomas@uconn.edu (M.S.T.); lindsey.huang@uconn.edu (L.H.); chelsea.garcia@uconn.edu (C.G.); junichi.sakaki@uconn.edu (J.R.S.); christopher.blesso@uconn.edu (C.N.B.); ock.chun@uconn.edu (O.K.C.)

**Keywords:** metabolic syndrome, eggs, plant-based diet, spinach, inflammation, oxidative stress

## Abstract

We recently reported that the inclusion of whole eggs in plant-based diets (PBD) increased plasma choline, lutein, and zeaxanthin in individuals with metabolic syndrome (MetS). The objective of the current study was to evaluate whether this dietary pattern would protect against oxidative stress and low-grade inflammation, two common characteristics of MetS. We recruited 24 men and women with MetS, who, after following a PBD for 2 weeks (baseline), were randomly allocated to consume either two whole eggs with 70 g of spinach/day (EGG) or the equivalent amount of egg substitute with spinach (SUB) as breakfast for 4 weeks. After a 3-week washout, they were allocated to the alternate breakfast. We measured biomarkers of oxidation and inflammation at baseline and at the end of each intervention. Tumor necrosis factor-alpha, interleukin-6, monocyte protein attractant-1, liver enzymes, and C-reactive protein, as well as total antioxidant capacity, paraoxonase-1 (PON-1) activity, and other biomarkers of oxidation were not different at the end of EGG or SUB or when compared to baseline. However, plasma malondialdehyde (MDA) concentrations were lower (*p* < 0.05) during the EGG and baseline compared to SUB. In addition, the increases in dietary lutein and zeaxanthin previously observed had a strong positive correlation with PON-1 activity (r = 0.522, *p* < 0.01) only during the EGG period, whereas plasma zeaxanthin was negatively correlated with MDA (r = −0.437, *p* < 0.01). The number of participants with MetS was reduced from 24 during screening to 21, 13, and 17 during the BL, EGG, and SUB periods, respectively, indicating that eggs were more effective in reversing the characteristics of MetS. These data suggest that adding eggs to a PBD does not detrimentally affect inflammation or oxidative stress; on the contrary, eggs seem to provide additional protection against the biomarkers that define MetS.

## 1. Introduction

Chronic low-grade inflammation and oxidative stress that often accompany metabolic syndrome (MetS) are significant factors causing the metabolic condition and associated pathophysiological consequences [1]. MetS, characterized by insulin resistance, hypertension, abdominal obesity, and dyslipidemia, triggers the alteration of cell signaling pathways, resulting in increased levels of inflammatory markers, lipid peroxides, and free radicals, causing cell damage and eventually leading to the clinical symptoms of the condition. Elevated products of oxidative stress as well as inflammatory markers, C-reactive protein (CRP) [2], tumor necrosis factor-alpha (TNF-α), and interleukin-6 (IL-6) [3] have been documented to contribute to the pathogenesis of MetS [4].

The dietary intake of antioxidant-rich foods can reduce the adverse effects of oxidative stress [5]. Plant-based diets (PBDs), through their antioxidant and anti-inflammatory properties, may reduce the development and progression of MetS [6]. There is convincing evidence that PBDs modulate immunological and inflammatory processes [7,8]. The effect may vary depending on the definition of vegetarianism and the individual nutrient components included in each [9]. Still, when compared to an omnivorous diet, vegetarian diets typically include a higher content of phytochemicals, antioxidant micronutrients such as vitamins C and E, fiber, and reduced saturated fats, which may contribute to a protective effect [10]. These beneficial dietary components and their metabolites help stabilize the gut microbiome, thus providing anti-inflammatory effects [11]. The vegetarian and vegan diets are associated with improved insulin sensitivity, reduced oxidative stress, and decreased concentrations of CRP [6,12].

Another natural source of antioxidants is whole eggs. In addition to their exceptional nutritional quality, eggs are rich in antioxidants such as vitamins A and E, selenium, lutein, and zeaxanthin [13]. The lipophilic antioxidants present in the egg yolk are highly bioavailable due to lipid content in the yolk. Eggs may provide natural sources of omega-3 fatty acids, vitamin B12, vitamin D, and bioavailable iron that are potentially deficient nutrients in a vegetarian diet. The anti-inflammatory effects of eggs observed previously in a carbohydrate-restricted diet [14] and after consumption of three eggs/d [15] suggest that eggs may complement the antioxidant effects of a PBD in treating MetS.

We have previously demonstrated the complementary effect of two whole eggs for 4 weeks to this protective diet, which reduced BMI and weight with corresponding increases in plasma HDL cholesterol (HDL-C), large HDL particles, and plasma choline, lutein, and zeaxanthin [16]. We reported the improvements in the dietary concentration of these nutrients and HDL-C after whole egg intake compared to egg substitutes in adults with MetS. In this study, as an extension of these findings, we investigated whether egg intake influenced oxidative stress and inflammation in these subjects. We hypothesized that whole egg consumption would not exacerbate the risk factors associated with MetS, but rather complement the PBD through its antioxidant properties as compared to egg substitutes.

## 2. Materials and Methods

### 2.1. Experimental Design

This randomized, controlled, cross-over designed dietary intervention was previously reported [16]. After screening to meet the criteria for MetS, participants were asked to follow a PBD (lactovegetarian) throughout the 13-week intervention. Participants were allowed to consume dairy, vegetables, fruits, and grains while restricting the consumption of meat, poultry, fish, and eggs other than those provided during the treatments. In addition, after a 2-week washout (without eggs and spinach), participants were randomly allocated to consume either two whole eggs per day (EGG) or the equivalent amount (1/2 c) of egg substitute with spinach (70 g) as an omelet for breakfast every day for 4 weeks. The egg substitute was egg whites from the Egg Beaters brand with zero cholesterol as described previously [16]. Following a 3-week washout, participants were assigned to the opposite treatment.

A registered dietitian advised the participants and ensured compliance with the diet. The intervention scheme was reported earlier [16]. Thirty participants (49.3 ± 8 y) classified with MetS according to National Cholesterol Education Program–Adult Treatment Panel III (NCEP-ATP III) guidelines [17] enrolled in the 13-week diet intervention. At the screening, each participant signed the written informed consent form. With 6 dropouts, 24 participants completed the intervention. This study was registered at Clinicaltrials.gov (protocol NCT04234334) and was approved by the University of Connecticut, Storrs Institutional Review Board under protocol H19-178.

### 2.2. Blood Sample Collection

Blood was drawn from MetS participants at baseline (BL) (week 2) and the end of each treatment (week 6 and week 13). After a 12 h overnight fast, blood samples were collected into EDTA coated vacutainers and were immediately centrifuged at 2000× *g* for 20 min at 4 °C for the separation of plasma. Blood collected in vacutainers without anticoagulant was left to clot at room temperature for 20 min before centrifuging to separate serum. Serum and plasma aliquots were stored at −80 °C until further analysis.

### 2.3. Liver Enzymes and C-Reactive Protein (CRP)

CRP and liver enzymes alanine transaminase (ALT) and aspartate transaminase (AST) were measured using an automated clinical chemistry analyzer (Cobas C 111, Roche Diagnostics, Indianapolis, IN, USA) as previously described [16,18].

### 2.4. Plasma Cytokines

Plasma tumor necrosis-α (TNF-α), monocyte protein attractant-1 (MCP-1), and interleukin-6 (IL-6) were measured following the manufacturer’s instructions using a commercially available sandwich enzyme-linked immunosorbent assay (ELISA) kit (Invitrogen, Carlsbad, CA, USA) and quantified using a BioTek Synergy microplate reader (BioTek Instruments, Inc., Winooski, VT, USA).

### 2.5. Total Antioxidant Capacity (TAC)

TAC was measured by assessing the extent of oxidation of the 2,2′-azino-di-3-ethylbenzthiazoline sulfonate (ABTS) radical. BioTek Synergy 2 Multi-Mode Microplate Reader with Gen5 Software (BioTek Instruments, Inc.) was used to measure absorbance [19].

### 2.6. Glycoprotein A (GlycA)

GlycA was measured spectroscopically by proton nuclear magnetic resonance (NMR) using the NMR LipoProfile^®^ spectra from plasma samples [20]. This clinical biomarker indicates the glycosylation levels of acute-phase proteins whose elevation reflects systemic inflammation [20].

### 2.7. Malondialdehyde (MDA), 8-Isoprostanes, and Oxidized LDL (oxLDL)

Plasma MDA and 8-isoprostanes were measured using commercially available ELISA kits from MyBioSource, San Diego, CA, USA and Cayman Chemical Company, Ann Arbor, MI, USA, respectively [21]. OxLDL was measured using a solid-phase capture sandwich ELISA from G Biosciences (St. Louis, MO, USA). Using plasma, reactions were carried out in 96-well plates and quantified using a microplate reader (BioTek Instruments, Inc., Winooski, VT, USA) [22].

### 2.8. HDL Components—Paraoxonase-1 (PON-1) Activity and Serum Amyloid A (SAA)

PON-1 lactonase activity toward delta-valerolactone (Sigma-Aldrich, Burlington, MA, USA) was determined [23,24] using a modification of the method described by Khersonsky and Tawfik [25]. According to the manufacturer’s instructions, SAA was measured in serum using a commercially available sandwich ELISA kit (Invitrogen, Carlsbad, CA, USA) and absorbance was read at 450 nm using a microplate reader (BioTek Instruments, Inc., Winooski, VT, USA) as previously reported [18].

### 2.9. Metabolic Syndrome Evaluation

We evaluated the parameters of the metabolic syndrome (waist circumference, blood pressure, plasma triglycerides, HDL-C, and fasting glucose) as previously reported [16]. We compared these values to those during screening to determine how many individuals resolved their MetS across the intervention during BL, EGG, and SUB periods.

### 2.10. Statistical Analysis

Statistical analysis was performed using SPSS version 17, statistical software for Windows (SPSS, Inc., Chicago, IL, USA). Significance was defined as *p* < 0.05 and data values were reported as mean ± standard deviation. A repeated-measures ANOVA evaluated differences over time (repeated measure) among BL, EGG, and SUB periods. Fisher’s test was used to detect significant differences among groups.

## 3. Results

### 3.1. Plasma CRP, Liver Enzymes, and Inflammatory Cytokines

The plasma concentration of CRP did not change for both EGG and SUB compared to BL. There were no significant changes in the concentration of TNF-α, MCP-1, and IL-6 throughout the intervention. Similarly, AST and ALT remained stable and within normal limits throughout the interventions. Data are shown in Table 1.

### 3.2. HDL Components—PON-1 and SAA

SAA concentrations were not significantly different throughout the 13-week intervention (BL: 5.91 ± 2.65 ng/mL, EGG: 6.25 ± 2.89 ng/mL, SUB: 6.17 ± 2.57 ng/mL), (Figure 1a). Similarly, no differences were seen in PON-1 activity after EGG (8.03 ± 2.7 U/mL) and SUB (8.04 ± 2.6 U/mL) when compared to BL (7.94 ± 2.8 U/mL) (Figure 1b). However, a strong positive correlation between PON-1 and dietary lutein and zeaxanthin (r = 0.522, *p* < 0.05) was observed during the EGG period (Figure 2). Previously, we have reported significant increases in dietary lutein and zeaxanthin after EGG (9190 ± 1527 µg) as well as SUB (9179 ± 2188 µg) when compared to BL (3151 ± 4382 µg) [16]. Dietary records were analyzed using the Nutrition Data System for Research (NDSR) (Nutrition Coordinating Center, University of Minnesota) [11].

### 3.3. Biomarkers of Oxidative Stress—TAC, MDA, 8-Isoprostanes, GlycA, and Oxidized LDL

Data are shown in Table 2. MDA concentrations were significantly decreased at BL and after EGG intake compared to SUB. There were no differences in the concentration of 8-isoprostanes, GlycA, and oxLDL between diets. We also found a negative correlation between MDA concentration and plasma zeaxanthin, as depicted in Figure 3. Plasma zeaxanthin values were previously reported [16], and they were significantly increased after EGG treatment (93.5 ± 50.8 nmol/L) compared to SUB (73.1 ± 38 nmol/L) and BL (68.6 ± 34.6 nmol/L).

Plasma TAC was also maintained throughout the intervention (BL: 183.98 ± 36.11 mg VCE/L, EGG: 185.87 ± 40.44 mg VCE/L, SUB: 172.25 ± 35.44 mg VCE/L), as shown in Figure 4a. We also found a negative correlation between oxLDL and TAC only during the EGG period, as shown in Figure 4b.

### 3.4. Changes in MetS Criteria

The effects of a PBD at baseline and during EGG and SUB periods in resolving the characteristics of the MetS is shown in Figure 5.

Of the 24 adults with MetS criteria enrolled in this study, 4 participants were no longer classified as having MetS after following a PBD for 2 weeks (BL) and 7 after SUB intake. However, EGG intake reversed the metabolic syndrome in 11 participants. Among subjects in the EGG group, seven participants were allocated to the EGG group first and four to the SUB group first. The individual components modified during each treatment were variable among participants.

## 4. Discussion

In this study, we found that daily intake of two eggs/day combined with spinach for 4 weeks while following a plant-based diet lowered the lipid peroxidation product MDA without increasing the markers of inflammation. We also found that the inclusion of whole eggs (EGG) helps to reverse MetS when compared to a plant-based diet (BL) or a plant-based diet including egg whites (SUB). These findings strengthen our hypothesis that eggs in combination with a lactovegetarian diet maintain the antioxidant status of a PBD and may reduce oxidative stress biomarkers, therefore preventing the progression of the metabolic conditions of MetS. This is supported by the significant findings reported earlier from this 13-week intervention, that two eggs/day complement a PBD by lowering body weight and BMI and improving HDL-C, especially the large HDL [16]. Egg intake improved the plasma concentrations of choline, lutein, and zeaxanthin without increasing plasma glucose or LDL, consistent with previous studies in MetS patients [26,27].

MetS features an increased pro-oxidative and pro-inflammation state [28]. The extent of oxidative stress is dependent on the severity of MetS. The imbalance in the antioxidative protection against damaging free radical accumulation triggers aging by damaging cellular functions, altering signaling pathways, and activating endothelial cell injury through inflammatory responses. The prevalence of MetS increases with age [29]. This relation by itself states the urgency of improving dietary intake of antioxidants in the MetS population to prevent age-related comorbidities. Adherence to the Mediterranean diet benefits MetS by reducing inflammatory and oxidative stress markers while improving insulin sensitivity [30]. Similar effects in MetS are shown with vegetarian and vegan diets compared to omnivorous diets [12,31,32]. However, a vegetarian diet puts one at risk for nutrient deficiencies in vitamin B12, vitamin D, calcium, iron, selenium, omega-3, and protein, which are crucial for vital functions [33]. The addition of eggs fills this gap, as evidenced by the dietary intake of our participants as described elsewhere [16].

Although MetS is associated with increased plasma CRP [2], our participants maintained lower concentrations throughout the intervention. Vegetarian and vegan diets are negatively correlated with CRP [12,32,34]. Unlike previous studies of MetS with eggs where inflammatory markers were reduced [14,15], we observed no changes in this study. The new and more reliable marker for systemic inflammation, GlycA [20], with comparatively superior reliability to CRP, also did not change throughout the intervention. No other effects were observed by including eggs or spinach in the diet. The Multi-Ethnic Study of Atherosclerosis (MESA) has depicted this relationship between reducing CRP and inflammatory cytokines after a fiber-rich vegetarian diet compared to a fat-rich, dietary fiber-devoid omnivore diet [35]. The PBD offers more dietary fiber, which may alter the microbiome and improve bacterial diversity [36]. The inflammation score in an energy-restricted 8-week nutritional intervention showed that the protein source in the diet matters in reducing cardiometabolic risk factors and was lower when consuming vegetarian proteins [37].

Low circulating HDL-C is a criterion for MetS and is a well-documented cardiometabolic risk factor. The vital proatherogenic function of HDL involves cholesterol efflux through reverse-cholesterol transport from peripheral tissue to the liver, thus modulating systemic inflammation. Systemic and vascular inflammation generated by disease conditions disrupts the proatherogenic effects of HDL, converting them to dysfunctional HDL [38]. The concentration of the inflammation-related marker SAA and antioxidant enzyme PON-1 carried by HDL are inversely related to each other and are used as indicators for HDL functionality [22,39]. In this study, the concentration of SAA and the lactonase activity of PON-1 were maintained throughout the intervention, even though an increase in HDL-C was observed after EGG treatment [16]. Antioxidant carotenoids, lutein, and zeaxanthin are mainly transported by HDL, which offers additional protective functions. We observed a strong positive correlation between PON-1 and dietary lutein and zeaxanthin only after EGG treatment, probably because eggs allow the lipid milieu to absorb these lipophilic compounds.

PON-1 has a critical role in promoting antioxidant properties by protecting LDL from lipid peroxidation. Lipid peroxidation refers to the oxidative degradation of lipid products, causing cellular damage by accumulating free radicals. Evidence attests that oxidative stress is increased in MetS due to the associated fat accumulation [40]. Insulin resistance, dyslipidemia, and abdominal obesity associated with MetS increase the production of free radicals, consequently raising MDA, the primary product of lipid peroxidation [41]. MDA indicates oxidative damage to cells and, thus, is a good biological marker for oxidative stress [42]. MDA values were higher after the SUB period than the EGG period and BL, indicating that eggs did not increase lipid peroxidation. The increases in HDL and corresponding increases in bioavailable carotenoids from egg yolk, both having antioxidant functions, may have contributed to this protective effect during the EGG period. The dietary antioxidants might have contributed to the total antioxidant capacity, leading to the negative correlation between TAC and plasma oxidized LDL concentrations.

The underlying conditions of MetS play a crucial role in the pathogenesis of atherosclerosis, causing a two-fold increase in disease risk for cardiovascular events [43]. Oxidation of LDL initiates atherosclerotic plaque formation. Oxidative stress silently mediates this progression. In this study, oxLDL was maintained throughout the intervention and egg consumption did not pose a risk by increasing oxLDL. Similarly, GlycA is a reliable cardiometabolic marker that can track the systemic acute phase responses, capturing inflammatory responses and providing information about progression to type 2 diabetes [44]. TMAO, the controversial atherogenic metabolite [45], was earlier shown to have no significant changes after egg consumption [16,27]. This evidence strongly suggests that consumption of two eggs/day did not detrimentally affect these key atherogenic risk factors of MetS. The maintenance of oxLDL, TAC, and 8-isoprostanes throughout the intervention must be due to the short duration of the diet. Long-term adherence (about 15 years) to a vegetarian diet causes lower oxidative stress, body weight, and cholesterol than omnivorous diets [46].

The first-line therapy for MetS is through diet modification, along with physical activity. This diet intervention motivated our participants to eat mindfully, even though their PBD diet was ad libitum and did not change their physical activity throughout the 13 weeks. This is evidenced by the overall improvement and reversal of MetS criteria during the diet periods. When compared to screening when all participants had at least three or more of the five criteria for MetS [17], the lactovegetarian diet and PBD for 2 weeks reversed MetS in four participants. When combined with two eggs/day for 4 weeks, this diet attenuated the symptoms and reversed MetS in 11 participants. After egg intake, most participants improved their plasma HDL, glucose, and blood pressure. Participants’ dietary intake indicated less carbohydrate and added sugar intake during the EGG period, which may have contributed to the wholesome and nutrient-rich PBD in reversing the MetS criteria.

This study’s findings strengthen our previous results, suggesting that a PBD with eggs is an effective strategy for treating MetS. We consistently found that the inclusion of whole eggs does not adversely affect the inflammatory status but reduces the oxidative stress, even with dietary cholesterol present, supporting eggs as an abundant source of antioxidants. The strength of this study is the 100% compliance to the PBD maintained by the participants throughout the intervention, even though all were habitual omnivores. The limitation of the study was the small sample size, the dropouts, and delays in the study due to the outbreak of COVID-19 and related circumstances. Additionally, 4 weeks might not have been long enough to observe a statistically significant change in biomarkers. Except for the parameters of MetS, no analyses were conducted at recruitment. This would have depicted the changes in MetS after 13 weeks of a PBD if measured. We assume the effects of a PBD throughout the intervention masked the impact of the eggs and the spinach that only helped maintain the biochemical status.

## 5. Conclusions

These results demonstrate that the inclusion of eggs in a PBD may provide beneficial effects in attenuating the symptoms of MetS by reducing the oxidative stress marker MDA compared to egg substitutes. Even with the cholesterol-rich egg yolk, two whole eggs did not increase inflammation in this at-risk population when consumed daily for 4 weeks, but reversed MetS criteria in a higher number of participants compared to baseline or the use of egg substitutes.

## Figures and Tables

**Figure 1 nutrients-14-02548-f001:**
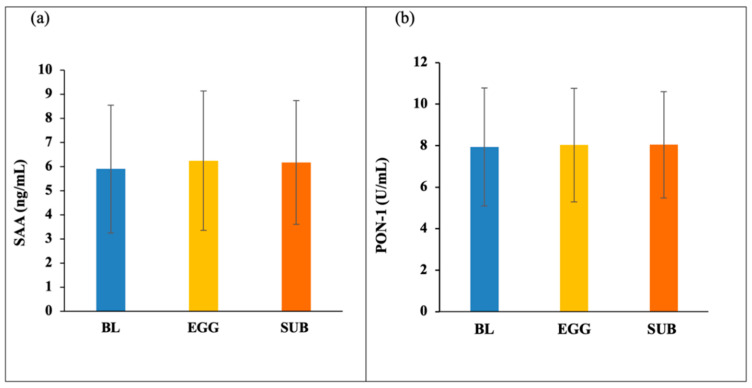
(**a**) Concentration of serum amyloid A (SAA) and (**b**) paraoxonase-1 (PON-1) measured at baseline (BL) and after EGG (whole eggs) and SUB (egg substitute) treatments for *n* = 24 subjects.

**Figure 2 nutrients-14-02548-f002:**
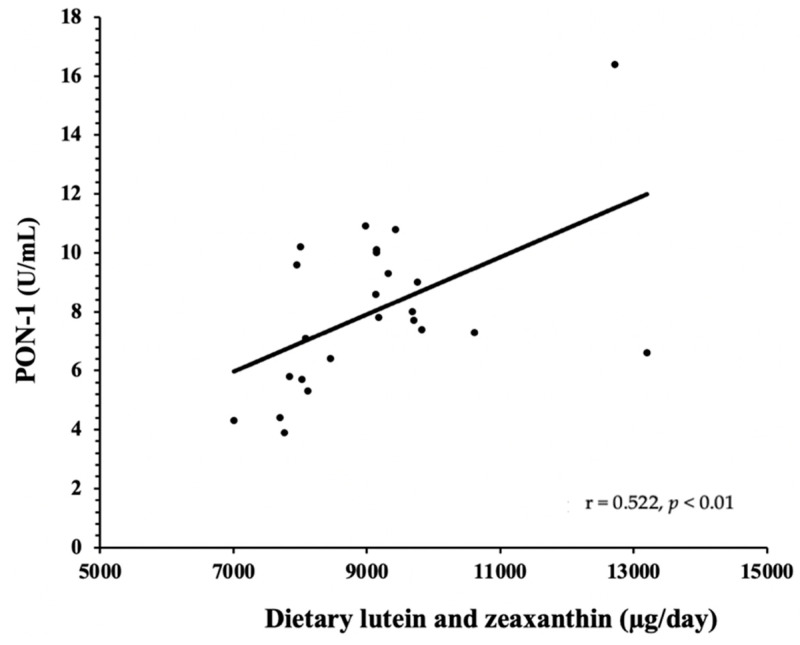
Dietary lutein and zeaxanthin correlated with PON-1, (r = 0.522, *p* < 0.01).

**Figure 3 nutrients-14-02548-f003:**
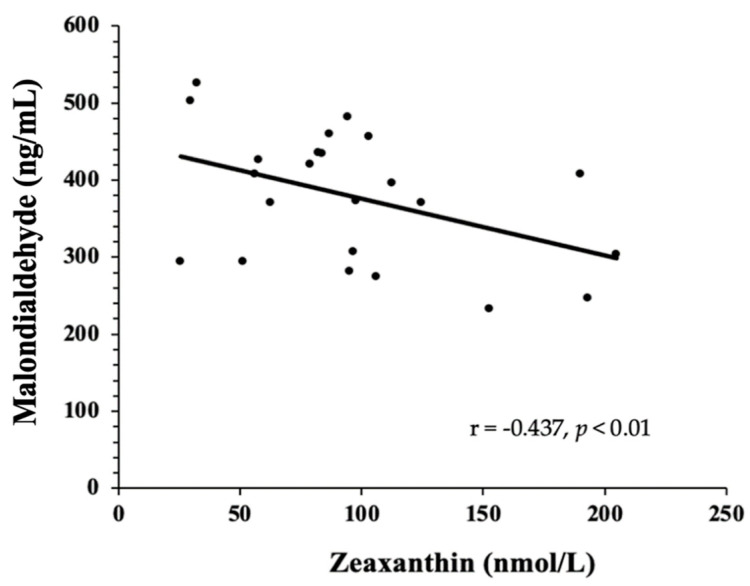
The concentratio of lipid peroxidation product, MDA negatively correlated with the concentration of plasma zeaxanthin after EGG intervention for *n* = 24 subjects.

**Figure 4 nutrients-14-02548-f004:**
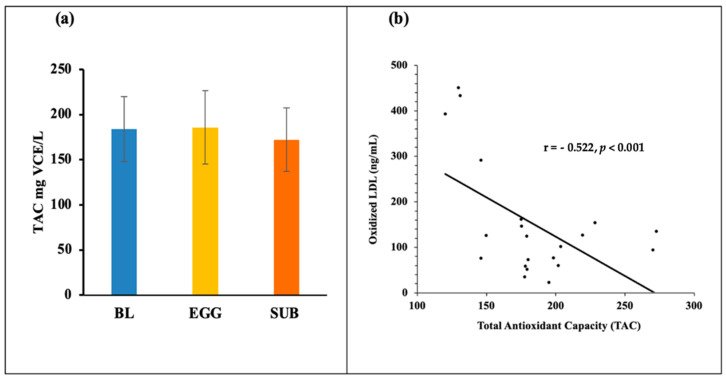
(**a**) The total antioxidant capacity at baseline (BL) and after EGG and SUB interventions. (**b**) TAC is inversely proportional to the concentration of plasma oxLDL after EGG intervention (r = −0.522, *p* < 0.001) for *n* = 24 subjects.

**Figure 5 nutrients-14-02548-f005:**
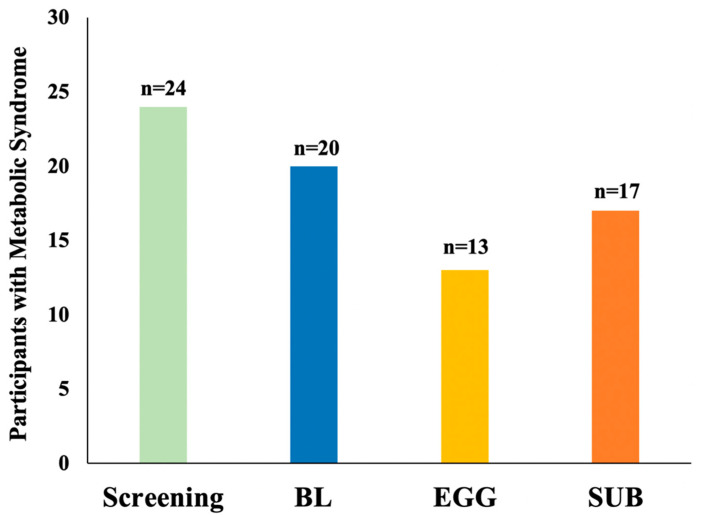
Participants classified with MetS at screening, at baseline (BL), and following the EGG and SUB periods.

**Table 1 nutrients-14-02548-t001:** Plasma concentrations of CRP, Liver Enzymes (ALT, AST), and cytokines (IL-6, MCP-1, TNF-α) at baseline (BL) and during the EGG (egg) and SUB (egg substitute) periods *.

Parameter	BL	EGG	SUB	*p* Value
CRP (mg/dL)	0.25 ± 0.24	0.40 ± 0.57	0.27 ± 0.26	0.448
ALT (U/L)	28.3 ± 17.0	28.7 ± 13.8	29.4 ± 21.3	0.679
AST (U/L)	23.0 ± 7.7	23.9 ± 8.2	22.4 ± 7.6	0.494
IL-6 (pg/mL)	3.5 ± 1.0	3.8 ± 1.2	3.7 ± 1.0	0.429
MCP-1 (pg/mL)	177.1 ± 65.4	174.1 ± 84.6	171.1 ± 73.8	0.632
TNF-α (pg/mL)	7.4 ± 1.5	7.3 ± 1.4	7.4 ± 1.7	0.855

* Data are presented as mean ± SD, (*n* = 24); CRP: C-reactive protein, ALT: alanine transaminase, AST: aspartate aminotransferase, IL-6: interleukin-6, MCP-1: monocyte chemoattractant protein-1, TNF-α: tumor necrosis factorα.

**Table 2 nutrients-14-02548-t002:** Biomarkers for measuring oxidative stress at baseline (BL) and the end of the EGG and SUB (egg substitute) periods *.

Parameter	BL	EGG	SUB	*p* Value
TAC (mg VCE/L)	184.0 ± 36.1	185.9 ± 40.4	172.3 ± 35.4	0.377
MDA (ng/mL) ^2^	397.1 ± 88.7 ^a^	389.2 ± 96.8 ^a^	426.2 ± 129.3 ^b^	0.049
8-Isoprostanes (pg/mL)	55.3 ± 15.2	58.4 ± 16.4	61.8 ± 11.8	0.277
GlycA (μm/L)	432.3 ± 66.5	434.4 ± 74.2	430.8 ± 66.3	0.908
OxLDL (ng/mL)	153.9 ± 119.1	151.1 ± 125.4	175.6 ± 117.5	0.057

* Data are presented as mean ± SD *n* = 24. ^2^ Values in the same row with different superscripts (^a,b^) are significantly different at a *p* < 0.05; total antioxidant capacity (TAC), malondialdehyde (MDA), 8-isoprostanes, glycoprotein A (GlycA), and oxidized LDL (OxLDL).

## Data Availability

Data for this study are available upon request from the principal investigator.

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
