# Peer review of "The Effects of Eggs in a Plant-Based Diet on Oxidative Stress and Inflammation in Metabolic Syndrome"

_nutrients, 2022, doi:10.3390/nu14122548_

Round 1
Reviewer 1 Report
Abstract is presented in a clear, succinct and direct way. Introduction is clear, containing all the necessary information to understand the objectives of this study. Methodology is clear and straightforward. Results and discussion are clear and in accordance with the objective.
Author Response
Abstract is presented in a clear, succinct and direct way. Introduction is clear, containing all the necessary information to understand the objectives of this study. Methodology is clear and straightforward. Results and discussion are clear and in accordance with the objective.
R. We thank the reviewer for his/her comments
Reviewer 2 Report
It is very interesting work. The title of the study needs some modification.
Author Response
Interesting work. The title of the study needs some modification.
Thanks for your kind comment. We would like to keep the title as it is since it is the one that better describes the study
Reviewer 3 Report
The Effects of Eggs in a Plant-Based Diet on Oxidative Stress and Inflammation in Metabolic Syndrome
The authors presented a study concerning the effects of combined vegetarian diet with egg consumption.
For the scientific value, I would have preferred a publication of the entire study and not the splitting into two papers with very similar content.
Did you performed a paired statistical analyze of the samples? So did you compared the samples of same participants in the egg and the control phase?
In eleven patients the EGG diet helped to reverse MetS and in seven the SUB diet. Are these partly the same participants after the cross-over? The group with the EGG period after the cross-over, already followed the vegetarian diet for 7 weeks instead of 2 weeks in the first EGG phase. Does this affects the results, so did you get more reversed MetS cases in the participants matched with the second EGG phase?
I’m very interested, if you found any correlations concerning this point. Because I think this is worth to discuss if a shorter vegetarian/egg diet is enough or if the 7 weeks longer period is necessary.
Figure 1:
- - is not levelled correctly and the scales are different
Figure 2:
- - Unit of PON-1 is missing
- - Which values were taken for the correlation in Figure 2? All values or just the values measured during the egg consumption?
- - How do you get different values for dietary lutein and zeaxanthin? I thought it is part of the egg. So where was it measured?
Figures general:
- - Figures are in different formats, font sizes and scales. Please update the appearance of the figures.
- - The number of biological replicates is missing in every figure.
Table 2:
- - When did you measured the values for MDA and the other factors in table 2? After the 4 weeks? Do you measured the values also in the follow-up during the 4 weeks?
Lower case C in line 243
Author Response
The Effects of Eggs in a Plant-Based Diet on Oxidative Stress and Inflammation in Metabolic Syndrome
- The authors presented a study concerning the effects of combined vegetarian diet with egg consumption.
- For the scientific value, I would have preferred a publication of the entire study and not the splitting into two papers with very similar content.
- We believe that the two papers have different content. We hypothesized that we would have very significant changes in oxidative stress and inflammatory markers following the egg intake.. We were expecting different results than what we got. This is the main reason why we separated into two papers but we do understand the points raised by the reviewer of consolidating into one paper.
- Did you performed a paired statistical analyze of the samples? So did you compared the samples of same participants in the egg and the control phase?
- We did a repeated measures ANOVA as indicated in the statistics section. Each subject is compared over time (the repeated sample) and the ANOVA is the three time points: Baseline, EGG and SUB.
- In eleven patients the EGG diet helped to reverse MetS and in seven the SUB diet. Are these partly the same participants after the cross-over? The group with the EGG period after the cross-over, already followed the vegetarian diet for 7 weeks instead of 2 weeks in the first EGG phase. Does this affects the results, so did you get more reversed MetS cases in the participants matched with the second EGG phase?
- I understand what the reviewer is saying. However, we started with 13 participants with EGG and 11 with SUB for these 24 participants. We just measured who no longer had metabolic syndrome at the end of each period. Therefore we believe that participants in EGG or SUB had about the same time whether it was 6 weeks (2 weeks baseline + 4 weeks of treatment) or 13 weeks (2 weeks baseline, 4 weeks first treatment and 3 weeks washout and 4 weeks second treatment) of being in the pant-based diet.
- I’m very interested, if you found any correlations concerning this point. Because I think this is worth to discuss if a shorter vegetarian/egg diet is enough or if the 7 weeks longer period is necessary.
The vegetarian diet was in fact for a total of 13 weeks. 2 weeks of washout, 4 weeks of EGG or SUB, 3 weeks of washout and 4 weeks of SUB or EGG. The difference that we wanted to show in the study what would happen if we added whole eggs or egg whites to the plant-based diet for 4 weeks.
Figure 1:
- - is not levelled correctly and the scales are different
- We have improved the presentation Figure 1. Thanks for checking on this
Figure 2:
- - Unit of PON-1 is missing
- - Which values were taken for the correlation in Figure 2? All values or just the values measured during the egg consumption?
- R. We have added the units for Figure 2. We only found correlations during the egg period. We found no correlations at baseline or at the end of the SUB period
- - How do you get different values for dietary lutein and zeaxanthin? I thought it is part of the egg. So where was it measured?
We did very detailed dietary records in our previous paper (reference 11) and we obtained the total amount of lutein and zeaxanthin that was consumed at baseline and after the EGG and SUB periods. This correlation is only with the EGG. We have added some clarification about this point.
Figures general:
- - Figures are in different formats, font sizes and scales. Please update the appearance of the figures.
- - The number of biological replicates is missing in every figure.
- R. We have tried to standardize the figures. We have added the number of biological replicates in the description of the figures as suggested by the reviewer.
Table 2:
- - When did you measured the values for MDA and the other factors in table 2? After the 4 weeks? Do you measured the values also in the follow-up during the 4 weeks?
The MDA values as well as all the values were measured at the end of baseline (2 weeks in the plant-based diet), at the end of the EGG and at the end of the SUB periods
Lower case C in line 243.
We have changed all the HDL-C to capital C since this is a more common terminology. Thanks for checking for consistency
Reviewer 4 Report
Line27: Mets must be MetS.
Statistical Analysis: Which Post Hoc Tests with ANOVA was used should be indicated. For ANOVA, although there is no significant difference, specific F and P values should be given.
Figure 2: It is unclear how the correlation between PON-1 and dietary lutein and zeaxanthin was generated. Importantly, how to determine the levels of dietary lutein and zeaxanthin should be indicated.
Figure 3: How about the correlation between MDA concentration and plasma zeaxanthin of after SUB intervention? How about the correlation of plasma zeaxanthin and PON-1 activity during the EGG period? For the plasma zeaxanthin and PON-1 activity, how to calibrate for potential influencing factors from weight, gender, age, etc.
The authors declare plasma malondialdehyde (MDA) concentrations were lower during the EGG and baseline compared to SUB. If MDA is one important factor, how to explain that EGG looks better than BL in slowing down the MetS?
Figure 5: This must be supported by reasonable statistical methods.
Author Response
Line27: Mets must be MetS.
- Changed accordingly
Statistical Analysis: Which Post Hoc Tests with ANOVA was used should be indicated. For ANOVA, although there is no significant difference, specific F and P values should be given.
The post-hoc test is now indicated in the methods section. We have added all P values in Tables 1 and 2 as suggested by the reviewer
Figure 2: It is unclear how the correlation between PON-1 and dietary lutein and zeaxanthin was generated. Importantly, how to determine the levels of dietary lutein and zeaxanthin should be indicated.
- We determined the levels of dietary lutein and zeaxanthin by using dietary records in our previous paper (reference 11). We have added a sentence for clarification (highlighted)
Figure 3: How about the correlation between MDA concentration and plasma zeaxanthin of after SUB intervention? How about the correlation of plasma zeaxanthin and PON-1 activity during the EGG period? For the plasma zeaxanthin and PON-1 activity, how to calibrate for potential influencing factors from weight, gender, age, etc.
We found no correlations at all between MDA and zeaxanthin during the SUB only with EGG. And no correlation between PON-I and zeaxanthin during the EGG. It was impossible to adjust for age, gender or weight with only 24 subjects.
The authors declare plasma malondialdehyde (MDA) concentrations were lower during the EGG and baseline compared to SUB. If MDA is one important factor, how to explain that EGG looks better than BL in slowing down the MetS?
- R. You mean lower than SUB right?. BL and EGG are not statistically significant. We were hoping to have more measurements being more significant with the eggs due to all the components in eggs that have antioxidative and anti-inflammatory properties, that was our initial hypothesis. So we were in fact disappointed that we only found differences in MDA
Figure 5: This must be supported by reasonable statistical methods.
- We just wanted to show the number of subjects that no longer had metabolic syndrome at the end of each intervention when compared to screening. These are just categorical values.
Round 2
Reviewer 3 Report
Dear Authors,
sorry that I bring up the termes of diet again. You said “We just measured who no longer had metabolic syndrome at the end of each period.” Can you tell me the number of participants who no longer had MetS after the first EGG period and the number after the second egg period? Is it the same? Even if you were focused on the question what would happen if whole eggs or egg whites are added to vegetarian diet for 4 weeks, you cannot ignore the effects of the vegetarian diet itself.
Author Response
The reviewer has a very valid point and sorry for not answsering properly to your question in our previous revision. Seven subjects started with EGG and 4 with SUB or the 11 subjects who no longer have metabolic syndrome. We have added a paragraph at the end of the figure to clarify this point (highlighted)
Reviewer 4 Report
The P value of 8-Isoprostanes in Table2 is worng (0,277).
Author Response
We have corrected the P value. Thanks for checking